# Multivariate time-series analysis of biomarkers from a dengue cohort offers new approaches for diagnosis and prognosis

**Baptiste Vasey[1], Anuraj H. Shankar[2], Bobby Brooke Herrera[3,4], Aniuska Becerra[5], Kris Xhaja[5], Marion Echenagucia[6], Sara R. Machado [7], Diana Caicedo[8], John Miller[9], Paolo Amedeo[9], Elena N. Naumova[10], Irene Bosch [3,11,12]\*, Norma Blumenfeld deBosch[5]**

1 Nuffield Department of Surgical Sciences, University of Oxford, Oxford, United Kingdom, 2 Nuffield Department of Medicine, Centre for Tropical Medicine and Global Health, University of Oxford, Oxford, United Kingdom, 3 E25Bio Inc., Cambridge, Massachusetts, United States of America, 4 Department of Immunology and Infectious Diseases, Harvard T.H. Chan School of Public Health, Boston, Massachusetts, United States of America, 5 Center for Infectious Diseases and Vaccine Research, University of Massachusetts, Worcester, Massachusetts, United States of America, 6 Centro Nacional de Hemofilia at Banco Municipal de Sangre, Universidad Central de Venezuela, Caracas, Venezuela, 7 Department of Health Policy, London School of Economics, London, United Kingdom, 8 Pontificia Universidad Javeriana, Cali, Colombia, 9 J. Craig Venter Institute, La Jolla, California, United States of America, 10 Friedman School of Nutrition Science and Policy, Tufts University, Boston, Massachusetts, United States of America, 11 Department of Medicine, Mount Sinai School of Medicine, New York, New York, United States of America, 12 Institute for Medical Engineering and Science, Massachusetts Institute of Technology, Cambridge, Massachusetts, United States of America

\* ibosch@e25bio.com

**Data Availability Statement:** All relevant data are within the manuscript and its supporting information files.

## Abstract

Dengue is a major public health problem worldwide with distinct clinical manifestations: an acute presentation (dengue fever, DF) similar to other febrile illnesses (OFI) and a more severe, life-threatening form (severe dengue, SD). Due to nonspecific clinical presentation during the early phase of dengue infection, differentiating DF from OFI has remained a challenge, and current methods to determine severity of dengue remain poor early predictors. We present a prospective clinical cohort study conducted in Caracas, Venezuela from 2001–2005, designed to determine whether clinical and hematological parameters could distinguish DF from OFI, and identify early prognostic biomarkers of SD. From 204 enrolled suspected dengue patients, there were 111 confirmed dengue cases. Piecewise mixed effects regression and nonparametric statistics were used to analyze longitudinal records. Decreased serum albumin and fibrinogen along with increased D-dimer, thrombin-anti-thrombin complex, activated partial thromboplastin time and thrombin time were prognostic of SD on the day of defervescence. In the febrile phase, the day-to-day rates of change in serum albumin and fibrinogen concentration, along with platelet counts, were significantly decreased in dengue patients compared to OFI, while the day-to-day rates of change of lymphocytes (%) and thrombin time were increased. In dengue patients, the absolute lymphocytes to neutrophils ratio showed specific temporal increase, enabling classification of dengue patients entering the critical phase with an area under the ROC curve of 0.79. Secondary dengue patients had elongation of Thrombin time compared to primary cases while the D-dimer formation (fibrinolysis marker) remained always lower for secondary compared

**Funding:** This work was supported by grant U01 AI45440 from the National Institutes of Health and funding from the Ministerio de Salud y Desarollo Social, Venezuela to IB and NB. BV was supported by a Mercator Fellowship and is currently a Berrow Foundation Lord Florey scholar at Lincoln College, Oxford. The funders had no role in study design, data collection and analysis, decision to publish, or preparation of the manuscript.

**Competing interests:** IB and BBH are co-founders of E25Bio Inc. (www.e25bio.com), a company that develops point-of-care diagnostics for fever-causing infectious agents.

to primary cases. Based on partial analysis of 31 viral complete genomes, a high frequency of C-to-T transitions located at the third codon position was observed, suggesting deamination events with five major hot spots of amino acid polymorphic sites outside in non-structural proteins. No association of severe outcome was statistically significant for any of the five major polymorphic sites found. This study offers an improved understanding of dengue hemostasis and a novel way of approaching dengue diagnosis and disease prognosis using piecewise mixed effect regression modeling. It also suggests that a better discrimination of the day of disease can improve the diagnostic and prognostic classification power of clinical variables using ROC curve analysis. The piecewise mixed effect regression model corroborated key early clinical determinants of disease, and offers a time-series approach for future vaccine and pathogenesis clinical studies.

## Author summary

Dengue fever results in a self-limiting, non-specific febrile illness. In approximately 10% of cases, the disease progresses to a severe, life-threatening syndrome. While hematological derangement is a key indicator of dengue, the mechanisms by which pathophysiological changes occur over the course of infection remain unclear. Additionally, there are limited clinical algorithms to facilitate rapid prognosis of dengue. We conducted a prospective clinical cohort study in Caracas, Venezuela to determine whether clinical and hematological parameters could distinguish dengue fever from other febrile illnesses, and identify early prognostic biomarkers of severe disease. Piecewise linear mixed effects regression models demonstrate that rates of change of albumin, fibrinogen, lymphocytes, platelets and thrombin time were significantly different between dengue and other febrile illnesses, and that the absolute value of albumin, fibrinogen, thrombin-antithrombin complex, thrombin time and partial thromboplastin time were prognostic of severe dengue on the day of defervescence. Our study offers extended insights into dengue pathogenesis and provides new approaches to dengue diagnosis and severity prognosis.

## Introduction

Dengue virus (DENV) is a mosquito-transmitted flavivirus that endemically circulates as four antigenically distinct serotypes (DENV1-4). Incidence has grown dramatically worldwide in recent decades, with an estimated 390 million annual infections, of which 96 million manifest clinically [1–3]. The most common clinical presentation is a self-limiting illness characterized by fever, headache, myalgia and arthralgia (dengue fever, DF) [4]. In approximately 10% of cases, the disease progresses to severe dengue (SD), characterized by increased vascular permeability resulting in capillary leakage, and can lead to lethal dengue shock syndrome (DSS) [4].

In 2000, DENV-3 re-emerged in Venezuela causing a prolonged, major outbreak with more than 83,000 reported cases [5]. We conducted a prospective clinical cohort study in Caracas between 2001–2005 on patients with suspected dengue to better understand the pathogenesis and progression of the disease with respect to immune cell activation and hemostasis. We aimed to correlate changes in selected blood biomarkers with clinical severity to identify diagnostic and prognostic biomarkers of DF and SD.

Dengue disease severity is affected by both viral and human immune factors [6, 7]. Primary exposure to any one of four DENV serotypes induces lasting immunity against reinfection by the same serotype [8–10]. However, epidemiological studies have shown an increased risk of

developing SD after secondary exposure to a different serotype [11, 12]. An antibody dependent enhancement hypothesis has been proposed wherein preexisting antibodies bind but do not neutralize virions of the subsequent infecting serotype [13]. Instead, these virus-immune complexes are recognized by Fcγ receptor-bearing cells that facilitate virus entry and replication, resulting in increased disease severity [14, 15]. Host protein cross-reactivity with DENV proteins has also been suggested to contribute to the pathogenesis of dengue [16–18].

Cytokine secretion and complement activation impair the endothelial barrier resulting in the leakage and loss of plasma components into the perivascular space, causing an increase in hematocrit and triggering the coagulation system. Studies on dengue patients indicate endothelial injury and/or activation by increased concentrations of von Willebrand factor (vWF) [19–22]. In addition, various processes contribute to thrombocytopenia including depression in bone marrow function, increase in megakaryocytes in the bone marrow, shortened platelet survival, and increased platelet consumption [23–27]. Neutropenia, lymphocytosis, as well as other elevated mediators including C3a, C5a, interleukins, tumor necrosis factor α, interferon γ, TRAIL, monocyte chemotactic protein 1, soluble suppression of tumorigenicity 2 (sST2, a member of the interleukin-1 receptor family), CXCL10 and histamine have also been associated with dengue pathobiology [28–40].

In SD, defects in coagulation such as prolonged activated partial thromboplastin time (aPTT), prothrombin time (PT), or thrombin time (TT) have been reported [41–43]. In addition, hyperfibrinolysis may occur in SD patients indicated by decreased plasma concentration of fibrinogen and increased concentration of its degradation products such as D-dimer (DD) [44–46]. A panel of peripheral white blood cell (WBC) and platelet counts, aPTT and PT, and blood chemistry including concentration of alanine aminotransferase (ALT) and aspartate aminotransferase (AST) was recently found to be predictive of SD diagnosis [47–49].

Current guidelines separate the course of the disease into three phases: (i) the febrile phase, from onset of symptoms to the day preceding defervescence, (ii) the critical phase, comprised of the day of defervescence and the two following days and (iii) the subsequent recovery phase [4, 50]. Whereas most studies in the field consider the day of onset of symptoms as day 1, we and others defined the day of defervescence as day 0 (D0) [51, 52]. This standardization of disease progression around a pivotal event allows better comparison of the physiological processes around the clinically important critical phase and permits a more robust statistical analysis as the variance due to different febrile phase length is eliminated. However, the practical clinical use of any predictors of DF or SD in this analytical context would require the means to estimate the day of disease at admission.

Due to nonspecific clinical manifestations during the early phase of dengue infection, differentiating DF from OFI has remained a challenge. Current laboratory-based methods to diagnose dengue include serological tests and virus antigen and nucleic acid detection, but their use is limited, especially in regions without access to sophisticated equipment and well-trained personnel. As there is currently no specific treatment for DF and SD, timely and proper hydration via intravenous fluid replacement remains the most effective way to reduce mortality, but requires early detection of patients at risk. However, the criteria commonly used to differentiate DF and SD, such as hemoconcentration and platelets count, remain poor predictors of SD [53]. In 2009, the World Health Organization [4] suggested new guidelines for the classification of dengue fever based on warning signs and evidence of plasma leakage [4]. Although the sensitivity of this new classification to detect cases of SD is high, its specificity remains low. Dengvaxia is the first FDA-approved vaccine for dengue; however, vaccination is limited to people with a prior history of dengue, as it can exacerbate disease [13, 54, 55]. A better understanding of the disease pathogenesis and the identification of early prognostic

biomarkers are therefore essential for the development of future diagnostic tests and supportive measures, and to define end-points for vaccination validation studies.

## Materials and methods

### Study population

Between 2001–2005, 204 patients with suspected dengue were enrolled from outpatient clinics within and around Caracas and referred to the Universidad Central de Venezuela, Centro Nacional de Hemofilia, Banco Municipal de Sangre. Inclusion criteria for the study were two or more days of febrile illness with one or more of the following symptoms: fever $\geq$38.5˚C, headache, myalgia, or a maculopapular rash, and no sign of other obvious infection (Fig 1B). Exclusion criteria were recumbent systolic blood pressure <85 mmHg, significant bleeding prior to study enrollment, and patients unlikely to attend or who did not attend follow up visits.

### Ethics statement

All adult patients provided written informed consent, and a parent or guardian of any child participant provided written informed consent on the child's behalf. The study was approved by the ethics review board of the Universidad Central de Venezuela, Centro Nacional de Hemofilia, Banco Municipal de Sangre (IRB number H-3693).

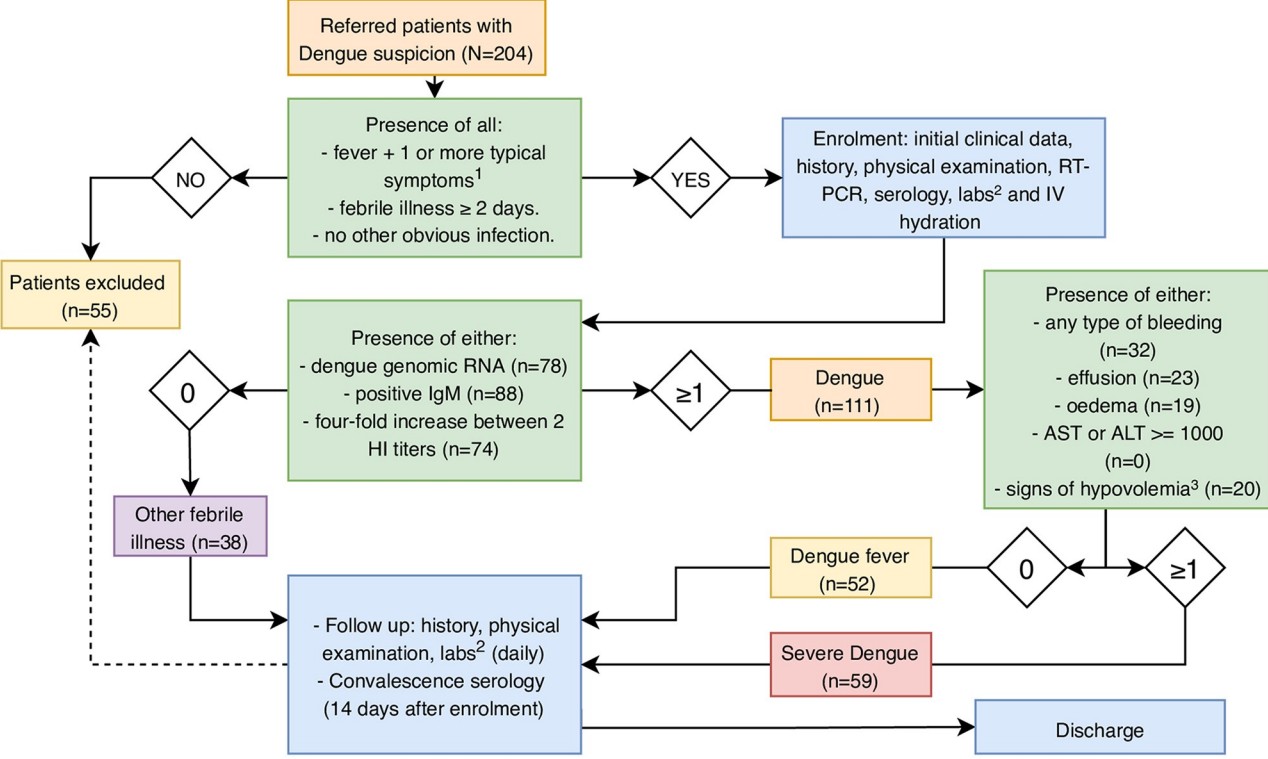

**Fig 1. A. Diagram of the time-line for data collection. B. Flow diagram representation of study inclusion and exclusion criteria.** Exclusion criteria included recumbent systolic blood pressure <85 mmHg, significant bleeding prior to study enrollment, and patients unlikely to attend or that did not attend follow-up visits. 149 patients completed the protocol. [1] Headache, myalgia or maculopapular rash; [2] Routine laboratory testing: complete blood counts, PT, PTT, TT, albumin, AST and ALT. Specialized coagulation studies: TAT, F1+2, fibrinogen, D-D and vWF; [3] Signs of hypovolemia were defined using the following criteria: (heart rate / systolic blood pressure) $\geq$ 1 or (systolic–diastolic blood pressure) $\leq$ 20 mmHg; HI: hemagglutination inhibition; RT-PCR: reverse transcription polymerase chain reaction.

## Data collection

Schematics of data collection are shown in Fig 1A. Briefly, after the initial visit and enrollment to the study, each patient attended the outpatient clinic daily. On every visit, medical history and physical findings, including hemorrhagic signs, were recorded on standardized case report forms and entered into a secured, centralized database. The patients also provided a venous blood specimen at each of these visits, prior to receiving IV saline solution. The blood samples were used to determine: the complete blood cell counts, the concentrations of albumin and liver enzymes and the activity parameters of coagulation and fibrinolysis. Patients were instructed to keep a record of oral temperatures three times a day and independently recorded these values at home. The patients were asked to return to the study clinic for a convalescent visit 2 weeks after enrollment for endpoint measurements as proxies for baseline status. Serological analyses for dengue immunoglobulin G and M (IgG/IgM) were performed on the day of study enrollment per protocol of the Instituto Nacional de Salud, 2 days after defervescence and on the convalescent visit.

sST2 concentrations were also measured in a randomly selected sample of 31 patients. One serum sample for each enrolled patient was processed for RT-PCR and a subset for exon sequencing. The archived data was utilized to generate multiple comparisons using MAFFT, with default settings. The consensus sequence was calculated using the default options of the EMBOSS Cons program (https://www.ebi.ac.uk/Tools/msa/emboss_cons/). The sequences analyzed are deposited at NCIB gene bank: FJ373304.1, FJ373303.1, FJ182015.1, EU854292.1, EU854291.1, EU660420.1, EU569691.1, EU569690.1, EU569689.1, EU569688.1, EU529691.1, EU529690.1, EU529689.1, EU529688.1, EU529687.1, EU529686.1, EU529685.1, EU529684.1, EU482614.1, EU482613.1, EU482612.1, KF955486.1, KF955487.1, KF955447.1, KF955330.1, KF955449.1, KF955450.1, KF955331.1, KF955452.1, KF955451.1, KF955453.1, KF955454.1. We confirmed that KF955487 and FJ182015 are two identical genome sequences from a duplicate of the same patient sample. The list contains a total of 31 unique DENV-3 genomes, out of 43 samples that were used in the analysis.

## Laboratory tests

Blood samples were collected in Vacutainer tubes (2% EDTA, 3.2% sodium citrate, and tubes without anticoagulant) and immediately used for analyses. Blood was tested for WBC and platelet counts, hemoglobin and hematocrit concentration (Beckman-Coulter). Reticulocytes were counted with a supravital stain. Plasma was used to perform coagulation tests including aPTT, PT, and TT, and concentration of fibrinogen was measured (Diagnostica Stago). Plasma concentrations of vWF (Diagostica Stago) prothrombin fragment 1+2 (F1+2, Dade-Behring), thrombin-antithrombin complex (TAT, Dade-Behring), DD (Diagnostica Stago), and sST2 were measured (MBL ELISA) according to the manufacturer's instructions. Concentrations of AST, ALT and albumin were measured in an automated analyzer (Wiener Lab, Argentina). Dengue IgM was measured by MAC ELISA and IgG titer with a hemagglutination inhibition assay (HI) from the Instituto Nacional de Salud in Venezuela [56]. Additionally, IgG ELISAs were utilized from commercially available sources (PanBio).

Dengue virus genomic RNA was isolated from febrile serum samples using the QIAmp Viral RNA kit (QIAGEN) and DENV serotype-specific reverse transcription and polymerase chain reaction (RT-PCR, QIAGEN) was performed as previously described [57]. The 31 sequenced viral genomes were uploaded to Genbank (S1 Table).

## Classification criteria

Patients were classified as dengue patients or as OFI based on the detection of genomic dengue RNA using RT-PCR, presence of IgM antibodies or seroconversion (four-fold increase in HI

levels in the second sample compared to the first). The HI levels were used to further classify dengue patients as primary infection (HI titer $\leq$ 1:1280) or secondary infection (HI titer $>$ 1:1280 in the first sample) [38].

The criteria used to classify a patient as having SD were genomically or serologically confirmed dengue infection plus at least one of the following: presence of any type of bleeding, presence of effusion or edema, signs of hypovolemia defined as pulse pressure (systolic blood pressure–diastolic blood pressure) $\leq$ 20 mmHg and/or heart rate $\geq$ systolic blood pressure. This corresponds to a risk-averse interpretation of the 2009 WHO criteria for SD, placing an emphasis on endothelial dysregulation.

## Statistical analysis

Statistical analyses were performed using R (v.3.5.0) and data completion using SAS 9.4. Two-sided Mann-Whitney U tests were performed due to the non-normal distributions of many variables, without correction for multiple comparisons. Statistics were only performed when at least five observations were present in each group for a given day and variable.

To examine longitudinal trends of outcome variables adjusted for patient-related covariates, we used piecewise linear mixed effects regression models as previously described [58–61]. The time period under observation was divided into two main stages: pre- (D-3 to D-1, $t_{time.before}$) and post- (D0 to D+3, $t_{time.after}$) defervescence. The equation for the piecewise regression model is as follows:

$$Y_{it} = \beta_0 + \beta_b * t_{time.before} + \beta_a * t_{time.after}$$

where $Y_{it}$ indicates longitudinal measurements for a variable of interest for $i$-patient at $t$-time; $t_{time.before}$ indicates the pre-defervescence day; $t_{time.after}$ indicates the post-defervescence day; and $\beta_0$ is the intercept of the model and a population estimate on the day of defervescence (D0). The regression parameters indicate the slopes, or rate of change per day for pre- ($\beta_b$) and post-defervescence ($\beta_a$) time period, utilizing D0 as the baseline. In this context, a statistically significant difference in slopes between two groups indicates a different day-to-day rate of change at patient level for a given biomarker. A statistically significant difference in the intercept indicates a difference in absolute value at D0 and at population level for a given biomarker.

We performed receiver operator characteristic (ROC) curves to determine whether knowledge about the day of illness could improve the DF/SD discrimination power of a particular variable. Optimal cutoffs were identified for each day between D-2 and D+1, for the indiscriminate first day of consultation and for the median value of each patient, using the Youden index (J = sensitivity + specificity -1). The areas under the curve (AUC) were calculated.

To perform initial alignment of DENV genomes, we used Prank (v.150803). The alignment was then submitted to codeml program in the PAML (v.4.8) suite of tools [62, 63]. From the codons output by codeml, separate alignments of first, second, and third codon positions were derived. For each column in each alignment, the changes from the ancestral sequence state to each other state were counted separately and divided by the number of isolates represented by nucleotides in the column to get a proportion for each mutation. Gap characters due to incomplete sequences were not counted. The graph was drawn using Plotnine, an implementation of ggplot in Python.

## Results

Of 204 patients enrolled, 149 completed the protocol. The median age was 20 years (range 5 to 74) and 61% were female. We confirmed dengue infection in 111 cases with 52 patients (47%)

**Table 1. Description of the study population**

| | Caracas Cohort | | | | |
|---|---|---|---|---|---|
| | Severe Dengue | Dengue Fever | Other Febrile Illnesses | Excluded | Total |
| **Total Group** | 59 | 52 | 38 | 55 | 204 |
| **Age Median (Q25-Q75)** | 18 (12–33.5) | 23.5 (14–37) | 23.5 (12.3–39.5) | 20 (14–35) | 20 (13–35) |
| **Age, <7 / ≥7** | 2/57 | 1/51 | 1/37 | 4/51 | 8/196 |
| **Sex, F / M** | 30/29 | 19/33 | 17/21 | 25/30 | 91/113 |
| **Serology, 1° / 2° dengue/Unknown** | 46/11/2 | 44/8/0 | NA | NA | NA |

classified as DF and 59 (53%) as SD. All received supportive therapy and survived. 90 (81%) patients experienced primary and 19 (17%) secondary dengue, while 2 (2%) could not be classified (Table 1). The patients were observed between 0 and 5 days (median = 1) before and between 1 and 7 days (median = 3) after defervescence (including defervescence day, D0). Overall, 70% of all daily designated clinical assessments and laboratory tests were completed, ranging from 3% to 97% completion. We determined that 65 out of 78 RT-PCR positive cases (83%) were infected with dengue serotype three.

## Non-parametric statistics

We performed comparative statistics on hematological test results obtained from patients with dengue (DF and SD) versus OFI. In the febrile phase, the dengue group had significantly decreased platelets and WBC median counts while having increased AST, ALT, DD and TAT complex median plasma concentrations. On D0, median plasma concentrations of fibrinogen and albumin as well as WBC absolute and neutrophils relative counts were decreased, whereas concentration of TAT, DD, AST, ALT, vWF and sST2 were elevated in the dengue group before defervescence compared to the OFI group. aPTT and TT were also prolonged in dengue cases compared to the OFI group. In the post-defervescence phase, dengue patients had decreased albumin and fibrinogen median plasma concentration, a decreased PT, an increased TT as well as elevated AST, ALT, vWF plasma concentration and relative lymphocytes count. (Figs 2A and 3A and S1 Fig).

When comparing patients with primary dengue to patients with secondary dengue, the albumin and hemoglobin concentration as well as the relative neutrophils and monocytes counts were significantly decreased on the day of defervescence. The WBC count and the relative lymphocytes count, representing lymphoproliferations, as well as the TT were elevated in secondary cases compared to primary cases (Fig 2A and S2 Fig).

On the day of defervescence (D0), SD patients had significantly decreased median plasma concentration of serum albumin and fibrinogen compared to DF, while DD plasma concentration was increased and the TT was elongated. The median plasma concentration of sST2 were elevated in the SD group between D-1 and D+1 (190.4 vs 152.6 at D-1, 256.4 vs 109.4 at D0 and 98.1 vs 64.5 at D+1), but the results were not statistically significant. (Figs 2A and 3C and S3 Fig). In the post-defervescence phase, the SD group had significantly decreased albumin and increased DD median plasma concentration compared to the DF group. SD patients had significantly higher heart rates and lower systolic and diastolic blood pressure around D0 compared to DF cases.

The full comparative statistics, including medians, quartiles and p-values for each variable, at each day and in each group can be found in S2 Table (DF vs OFI), S3 Table (primary vs secondary dengue) and S4 Table (DF vs SD).

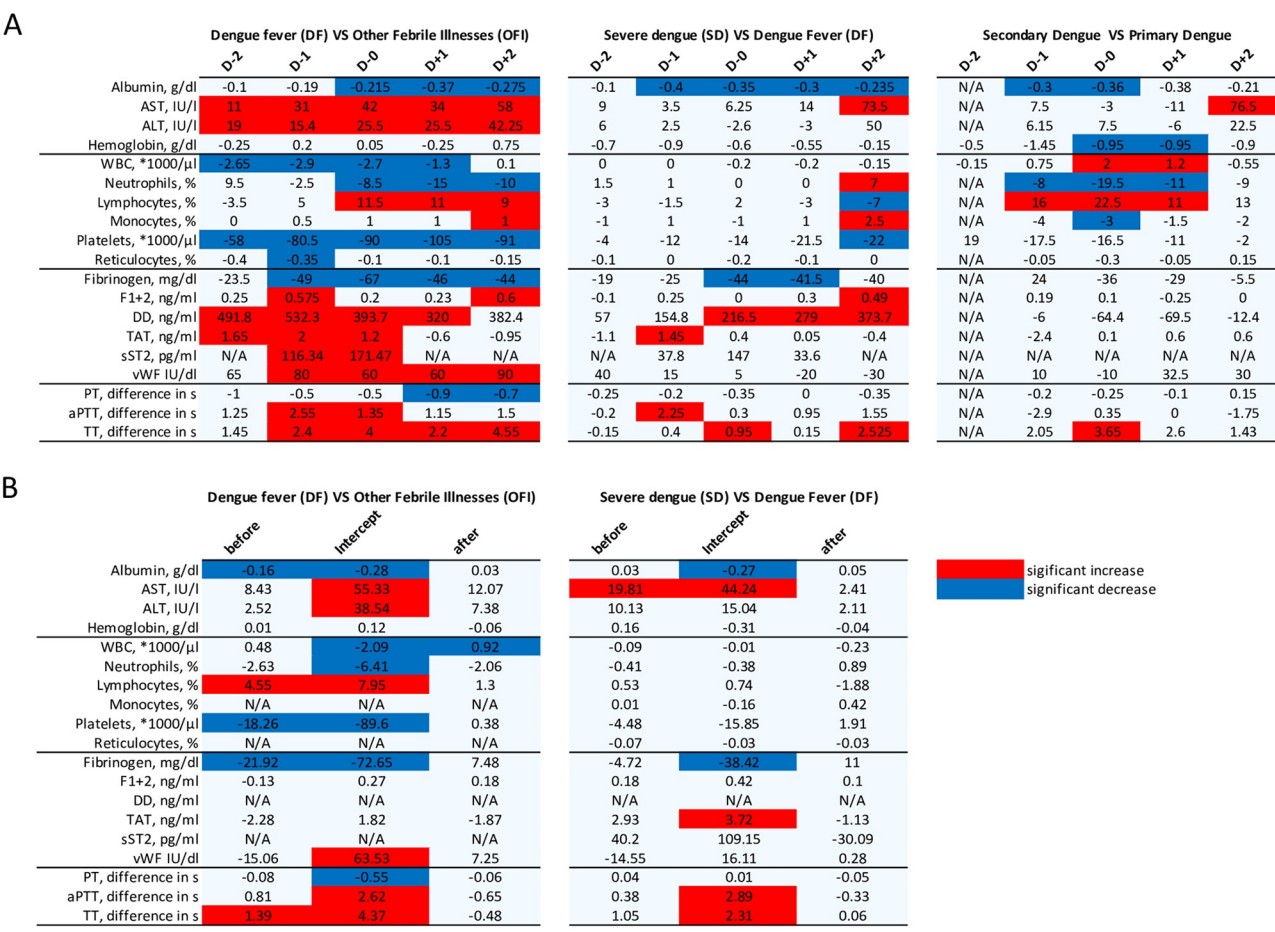

**Fig 2. Overview of the blood biomarkers changes between the different groups.** A. Significant changes in blood markers for each day using Mann-Whitney U test. *D0* represent the day of defervescence. In the comparison *dengue VS other febrile illnesses*, dengue comprises both dengue fever and severe dengue cases. B. Significant changes in blood biomarkers kinetic and intercept using the piecewise linear mixed effect regression model. *Before* and *after* represent two slopes while *intercept* is the biomarker's value on D0. Blue, significant decrease. Red, significant increase.

## Piecewise linear mixed effects regression

We next performed a piecewise linear mixed effects model analysis to investigate the evolution of each variable over time (fixed effect) and its expected value at D0 (intercept). Compared to OFI, dengue cases had a sharper increase in relative lymphocyte count and TT and a sharper decrease in platelet count during the pre-defervescence phase. Plasma concentrations of albumin and fibrinogen decreased in dengue patients while increased in OFI. On D0, the dengue group had lower albumin and fibrinogen plasma concentrations, as well as lower WBC, neutrophil and platelets counts. Plasma concentrations of AST, ALT and vWF as well as relative lymphocytes count, aPTT and TT were higher in the dengue group. During the post-defervescence phase, WBC counts increased in dengue patients while decreased in OFI cases (Figs 2B and 3B and S4 Fig).

The linear mixed effects models demonstrated that, compared to DF cases, SD cases had a steeper increase (slope is statistically different) in plasma concentration of AST during the febrile phase. On D0, SD patients had lower plasma concentrations of albumin and fibrinogen, serum concentrations of AST and TAT were higher, and prolonged aPTT and TT were

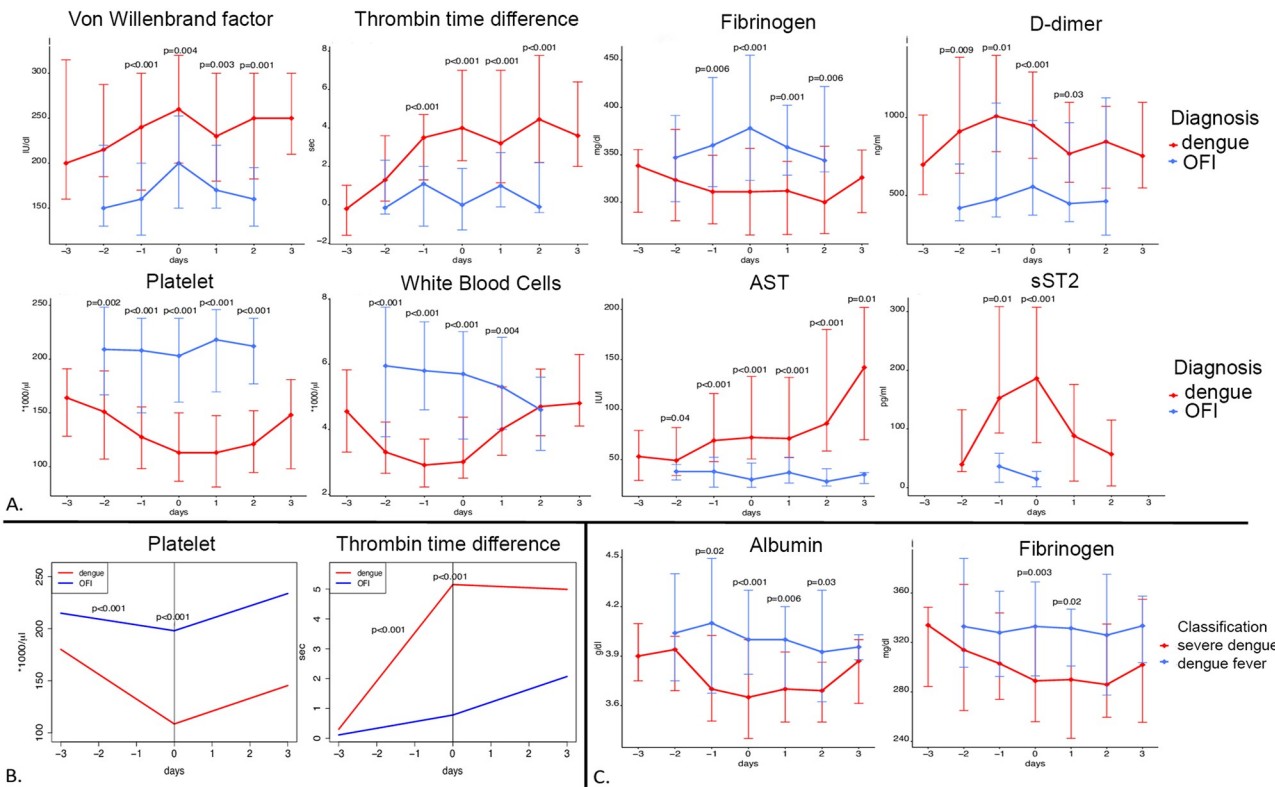

**Fig 3. Evolution of selected blood biomarkers for selected groups comparison over the course of disease.** A. Evolution of the medians (and interquartile ranges) with comparison between dengue and other febrile illness patients. B. Evolution of the biomarkers at patient level using piecewise linear mixed effects models with comparison between dengue and other febrile illness patients. C. Evolution of the medians (and interquartile ranges) with comparison between dengue fever and severe dengue patients. In A and C, the linking lines are displayed for a better visualization but do not represent the evolution of the biomarkers at patient level. D0 is the day of defervescence.

observed (Fig 2B and S5 Fig). No statistically significant differences were observed during the post-defervescence phase. The model could not be applied to some clinical variables like DD due to a relative lack of data points compared to the total number of random effects. The full model statistics can be found in S5 Table (dengue vs OFI) and S6 Table (DF vs SD).

## ROC statistics

We further conducted ROC analysis using the longitudinal clinical and hematological data to evaluate diagnostic and prognostic test value for candidate biomarkers. Counts of WBC (AUC = 0.89 at D-1), reticulocytes (AUC = 0.77 at D-1), and platelets (AUC = 0.88 at D+1), TT (AUC = 0.85 at D0), and plasma concentrations of AST (AUC = 0.82 at D0), ALT (AUC = 0.86 at D-2), vWF (AUC = 0.77 at D-1), fibrinogen (AUC = 0.75 at D0), TAT (AUC = 0.75 at D-1) and D-dimer (AUC = 0.80 at D-2) appeared as the best candidate biomarkers to differentiate between dengue and OFI. Plasma concentrations of albumin (AUC = 0.71 at D0), fibrinogen (AUC = 0.69 at D0) and D-dimer (AUC = 0.70 at D0) could be useful in distinguishing between DF and SD. The full tables with AUC, best cutoffs, sensitivity, specificity and confidence interval are accessible in S7 Table (DF vs OFI) and S8 Table (DF vs SD). We noticed large variation and difference of up to 0.21 in AUC when comparing the ROC of a biomarker at a defined day of illness to the ROC at the indiscriminate day of first

**Table 2. Evolution of the receiver operating characteristics statistics for selected biomarkers over the course of disease.** Classification between the dengue and OFI groups. First contact: day of first contact with healthcare (independently of the day of fever). Median days -2 to 1: a posteriori calculation of the biomarker's median value between D-2 and D+1. AUC: area under the curve; CI: confidence interval.

| | | Day | | | | | |
|---|---|---|---|---|---|---|---|
| Variables | | First Contact | -2 | -1 | 0 | 1 | Median days -2 to 1 |
| **White blood cells, \*1000/μl** | Best cutoff, Youden index | 5.25 | 5.55 | 3.95 | 3.35 | 4.25 | 4.08 |
| | AUC | 0.76 | 0.82 | 0.89 | 0.79 | 0.67 | 0.8 |
| | 95% CI AUC | 0.66–0.86 | 0.70–0.94 | 0.82–0.96 | 0.7–0.87 | 0.56–0.77 | 0.72–0.89 |
| **Neutrophils, %** | Best cutoff, Youden index | 84.5 | 57.5 | 72 | 51.5 | 47.5 | 51.75 |
| | AUC | 0.5 | 0.6 | 0.58 | 0.63 | 0.7 | 0.66 |
| | 95% CI AUC | 0.38–0.61 | 0.37–0.83 | 0.44–0.72 | 0.52–0.74 | 0.6–0.8 | 0.56–0.76 |
| **Lymphocytes, %** | Best cutoff, Youden index | 11.5 | 36.5 | 25.5 | 33.5 | 51.5 | 38.25 |
| | AUC | 0.51 | 0.61 | 0.57 | 0.65 | 0.72 | 0.67 |
| | 95% CI AUC | 0.40–0.63 | 0.39–0.82 | 0.43–0.71 | 0.54–0.76 | 0.62–0.82 | 0.57–0.77 |
| **Platelets, \*1000/μl** | Best cutoff, Youden index | 194.5 | 199.5 | 170 | 157.5 | 166.5 | 160 |
| | AUC | 0.74 | 0.76 | 0.81 | 0.85 | 0.88 | 0.84 |
| | 95% CI AUC | 0.64–0.83 | 0.61–0.90 | 0.7–0.92 | 0.78–0.92 | 0.81–0.95 | 0.76–0.92 |
| **Thrombin Time, difference in sec** | Best cutoff, Youden index | 3.3 | 0.1 | 2.45 | 2.8 | 1.95 | 1.75 |
| | AUC | 0.7 | 0.69 | 0.73 | 0.85 | 0.71 | 0.8 |
| | 95% CI AUC | 0.59–0.80 | 0.49–0.88 | 0.62–0.85 | 0.77–0.92 | 0.6–0.82 | 0.72–0.88 |

contact with healthcare (mostly between D-3 and D0 (Table 2). Nevertheless, none of these differences were statistically significant.

The above results suggest that better knowledge of the day of illness could improve the classification power of certain tests and provide clinicians with valuable insights about the onset of the critical phase. We therefore assessed the ability of biomarkers to discern the day of illness and identified the absolute lymphocytes/neutrophils ratio as a potential time predictor in dengue patients. In patients with confirmed dengue infection, there is a quasi-linear correlation between days (D-2 to D+1) and the proposed ratio (Fig 4). ROC statistics and optimal cutoffs were calculated to differentiate the critical day (D0) from previous days of illness in febrile dengue patients. The AUC was 0.79 and the best cutoff using the Youden index was 0.537 with 65% specificity and 80% sensitivity. Using a cutoff of 0.978, patients entering the critical phase could be identified with 90% specificity and 47% sensitivity.

## Genome analysis

In the 31 sequenced DENV3 genomes out of 78 RT-PCR positives (40%), we observed 49 amino acid mutations well distributed across the genome. We observed higher frequency of T-to-C/C-to-T substitutions than other mutations, and these transitions were 7.6-fold higher at the third codon position (S6A Fig). The limited number of collected specimens with completed sequence data precluded formal assessment of the association between mutations and disease severity. The four serotypes of dengue co-circulated during the dengue outbreak according to RT-PCR data, where dengue 3 represented the majority (83%). All the sequenced DENV-3 genomes were classified as Genotype V (S6B Fig).

## Discussion

Our findings demonstrate detailed differences in laboratory features between OFI, DF, and SD during a dengue epidemic in Caracas, Venezuela. The results of the mixed effects models were

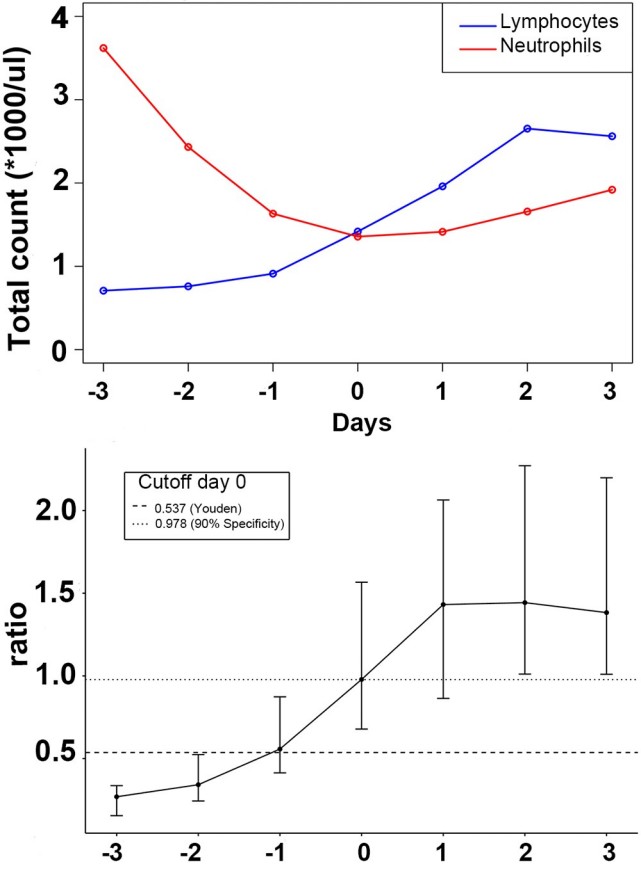

**Fig 4. Evolution of neutrophils and lymphocytes absolute counts and their ratio.** A. Evolution of the neutrophils and lymphocytes absolute counts' medians for dengue patients. B. Evolution of the neutrophils over lymphocytes ratio and selected cutoffs (method used) to differentiate between D0 and any previous day. The intervals represent the range between the first and the third quartiles. (- - -) sensitivity = 0.80, specificity = 0.65; (·····) sensitivity = 0.47, specificity = 0.90.

consistent with the results obtained using non-parametric statistics. At D0, 12 different laboratory parameters had classifier effects for dengue vs OFI and 6 when comparing DF with SD. The results of the piecewise mixed effects model indicate statistically significant differences in the rates of change of 5 biomarkers before, and one after, defervescence in dengue vs OFI and of one single biomarker in the pre-defervescence phase for DF vs SD.

At D0, coagulation biomarkers were important differentiators between DF and OFI, including increased TT, increased concentrations of vWF and TAT complex, and decreased concentrations of fibrinogen and DD. Additional diagnostic biomarkers were decreased concentration of albumin, increased concentration of sST2, decreased WBC and platelets counts, increased percentage of lymphocytes, and decreased percentage of neutrophils. Most of these biomarkers were also diagnostic during the febrile phase.

The results highlighted seven hematological biomarkers (albumin, AST, fibrinogen, DD and TAT-complex concentrations as well as aPTT and TT coagulation tests) as prognostic of SD at D0. These findings suggest an exacerbation of the dengue-related coagulation disorders in patients experiencing SD. The majority of our findings are consistent with the existing literature indicating coagulation factors can be used to define the severity of the disease. One

noticeable exception is the hematocrit, which was found to be lower in SD patients compared to DF, where it would be expected to be higher due to hemoconcentration. This could be explained by the hydration protocol in place during this hospital study. Another novel finding is that fibrinolysis is not as prominent in secondary dengue cases compared to primary cases as the concentration if DD was consistently lower compared to primary infections. This could be due to the more prominent extravasation of fibrinogen and lower serum levels, as well as a possible decrease in the hepatic synthesis of fibrinogen during secondary dengue infections. Notably, the differences in DD between primary and secondary were not statistically significant; rather, the results observed are general trends.

Some of the observations derived from our time-series analysis provide a better understanding of dengue pathogenesis. The elongation in TT and aPTT, coupled with decreased levels of fibrinogen and increased DD suggest a procoagulatory state with an associated activation of fibrinolysis as a reaction mechanism. This hypothesis is further supported by elevated TAT complex and F1+2 concentrations, suggesting an enhanced conversion rate from prothrombin to thrombin. Endothelial damage, as demonstrated by the elevated levels of vWF, is likely to be a trigger for the procoagulatory state alongside elevated tissue factor (TF) whose plasma levels have been shown to increase during febrile phases [22, 64]. The decreased levels of fibrinogen could be explained by an increased consumption due to the procoagulatory state but also by the formation of immune complexes with antibodies against viral proteins (e.g. anti-NS1). We therefore support the use of blood coagulation biomarkers (such as coagulation times, Fibrinogen, TAT complex, vWF, and DD), often not considered, to define severity end points in future dengue studies.

We observed that the timing of neutropenia was associated with platelet decrease. A possible explanation for the loss of neutrophils from the circulation during the febrile phase is neutrophil attachment, rolling, and extravasation, potentially mediated by increased expression of P-selectin in the vascular endothelium. This hypothesis is supported by an increase of P-selection mRNA in endothelial cells infected with dengue virus in vitro [65]. Clinical data in humans reporting endothelium activation and neutrophil extravasation during neutropenia further support this hypothesis [66]. To test this hypothesis, it would be useful to quantitatively evaluate the loss of neutrophils due to extravasation versus attrition or suppressed production as previously described in animal models [67].

To date, hematological studies have been primarily focused on dengue patients from Asia. Our study is the first to provide detailed information about the timing of hemostatic biomarker changes at each time point in a large dengue cohort in Latin America. In addition, we examined longitudinal trends of outcome variables in patients by uniquely utilizing piecewise linear mixed effects regression models. The rates of change of albumin, fibrinogen, lymphocytes, platelets and TT were shown to be significantly different between dengue and OFI patients in the febrile phase. From a clinical perspective, these findings suggest that at least two consecutive days of consultation and laboratory tests could increase the diagnostic power of several biomarkers.

Given the variation in accuracy of many biomarkers over the course of disease and the importance of close monitoring during the critical phase, we hypothesized that additional information about the day of illness relative to D0 would benefit clinicians and patients. This could improve the diagnostic and prognostic accuracy of specific biomarkers while simultaneously improving risk and patient flow management. Given a day of first contact with the healthcare system approximately evenly distributed between D-3 and D0 and a febrile phase lasting between 2 and 7 days, any prediction about the onset of the critical phase based only on the day of first symptoms or of first consultation remains unreliable. In our cohort, the ratio of decreasing neutrophils over increasing lymphocytes (absolute counts) was progressively more

pronounced over the course of the disease for patients infected with dengue virus. We propose this ratio as a surrogate to identify the day of illness relative to D0. A ratio over 1.0 classifies febrile patients entering the critical phase with a specificity of 90%. In resource limited settings, a better anticipation of the onset of the critical phase could help manage follow-up consultation and inpatient admissions. Moreover, the costs for such additional inference is minimal as the needed biomarker data are already routinely collected as part of the care process. The clinical use of this ratio would need confirmation in additional, larger cohorts.

The higher frequency of C-to-T/T-to-C substitutions at the third codon position observed in the genome analysis is consistent with our previously published studies on Zika and Ebola viruses [68, 69]. These data suggest that host RNA editing enzymes (e.g. ADARs) and chemical deamination may contribute to viral RNA diversity; however, further investigation is required to determine the exact mechanisms of the observed deaminations. Our phylogenetic analysis suggested that the epidemic DENV-3 strain originated in the Caribbean in 1998 (S6B Fig).

Our study has some limitations. Most of the patients in our cohort were infected with DENV-3 and generalization of our results would need confirmation in populations infected with other DENV serotypes. Due to the outpatient care setting of the study, patients presenting with more than 48 hours of fever, significant bleeding (based on clinical history and physician judgment) or signs of shock during the first visit, were excluded. This might have influenced the recruitment of SD patients. Moreover, reliable information about underlying comorbidities, which could explain abnormal biomarker levels, were available as per clinical history but not confirmed with other tests outside the protocol established in the study. Due to the requirement of a long 4 kb open reading frame for RT-PCR procedures (Schmidt et al, 2011), of the 43 samples submitted for sequencing at genomic facility, 25 resulted in complete and 6 in incomplete genomes, for a total of 31 unique genome sequences for dengue serotype 3. This represented 72% of the total number of samples submitted but only 40% of the 78 PCR positive patient samples acquired. The techniques for sequencing RNA viral genomes at that time resulted in a loss of some full genomic sequences and thereby precluded inference about the association between point mutations and disease severity. We provide the successfully sequenced genomes and their corresponding disease outcome in S1 Table to support findings of genome variability among patients' samples from the same outbreak.

The combination of classic and novel time-series statistics confirms and expands existing knowledge, offers extended insights into dengue pathogenesis, and provides possible new approaches to dengue diagnosis and severity prognosis that should have applicability in future studies, including those for vaccine trials.

## Supporting information

**S1 Table. Infecting viral genomes and associated disease outcomes.** DF: dengue fever; SD: severe dengue.
(XLSX)

**S2 Table. Difference in medians between the dengue and other febrile illness (OFI) groups.**
(XLSX)

**S3 Table. Difference in medians between the primary dengue (1˚) and secondary dengue (2˚) groups.**
(XLSX)

**S4 Table. Difference in medians between the severe dengue (SD) and dengue fever [68] groups.**
(XLSX)

**S5 Table. Piecewise linear mixed effects regression statistics with comparison between the dengue and other febrile illness (OFI) groups.** β0: intercept and value at D0; βb: slope time before [D-3 to D0]; βa: slope time after [D0 to D+3]; SD: standard deviation; 1s-t-test: 1 sample t-test; 2s-t-test: 2 samples t-test.
(XLSX)

**S6 Table. Piecewise linear mixed effects regression statistics with comparison between the dengue fever and severe dengue groups.** β0: intercept and value at D0; βb: slope time before [D-3 to D0]; βa: slope time after [D0 to D+3]; SD: standard deviation; 1s-t-test: 1 sample t-test; 2s-t-test: 2 samples t-test.
(XLSX)

**S7 Table. Evolution of the receiver operating characteristics statistics over the course of disease and comparison of the biomarkers' classification power between the dengue and other febrile illness groups.**
(XLSX)

**S8 Table. Evolution of the receiver operating characteristics statistics over the course of disease and comparison of the biomarkers' classification power between the dengue fever and severe dengue groups.**
(XLSX)

**S1 Fig. Evolution of all the analyzed blood biomarkers' medians with comparison between dengue and other febrile illness patients over the course of disease. The lower and upper error bars represent the first and third quartiles respectively.** The linking lines are displayed for a better visualization but do not represent the evolution of the biomarkers at patient level. *p-value ≤ 0.05; **p-value ≤ 0.01; ***p-value ≤ 0.001.
(TIF)

**S2 Fig. Evolution of all the blood biomarkers' medians with comparison between primary and secondary dengue patients over the course of disease.** The lower and upper error bars represent the first and third quartiles respectively. The linking lines are displayed for a better visualization but do not represent the evolution of the biomarkers at patient level. *p-value ≤ 0.05; **p-value ≤ 0.01; ***p-value ≤ 0.001.
(TIF)

**S3 Fig. Evolution of all the blood biomarkers' medians with comparison between dengue fever and severe dengue patients over the course of disease.** The lower and upper error bars represent the first and third quartiles respectively. The linking lines are displayed for a better visualization but do not represent the evolution of the biomarkers at patient level. *p-value ≤ 0.05; **p-value ≤ 0.01; ***p-value ≤ 0.001.
(TIF)

**S4 Fig. Evolution of the blood biomarkers at patient level using piecewise linear mixed effects models with comparison between dengue and other febrile illness patients.** No results displayed for D-dimer, monocytes, reticulocytes and sST2 as the models could partially not be fitted. *p-value ≤ 0.05; **p-value ≤ 0.01; ***p-value ≤ 0.001.
(TIF)

**S5 Fig. Evolution of the blood biomarkers at patient level using piecewise linear mixed effects models with comparison between dengue fever and severe dengue patients.** No

results displayed for D-dimer as the models could partially not be fitted. $^{*}$p-value $\leq$ 0.05; $^{**}$p-value $\leq$ 0.01; $^{***}$p-value $\leq$ 0.001.
(TIF)

**S6 Fig. Genome analysis on 31 DENV-3 isolates from patients experiencing dengue fever and severe dengue.** A. Distribution of amino acid mutations across the DENV-3 genome, separated by codon positions. B. Phylogenetic tree showing Caracas samples collected in 2001 in red, other Caracas samples in blue, and isolates from nearby locations in grey. Tree was found using RAxML rapid bootstrapping with 100 bootstrap replicates.
(TIFF)

## Acknowledgments

BV thanks AHS and IB for their supervision and mentoring. The authors are grateful to Lee Gehrke at the Massachusetts Institute of Technology for support of the work of BV and IB. To Alan Rothman and Francis Ennis for their help in protocol design and support to IB and NB. The authors thank Ana Aleman, Mariella Lilue, Carmen Carpio and Jose Angel Ilarraza for patient enrollment and data collection; the personnel of Instituto Nacional de Higiene Rafael Rangel for support on serologic testing; Isabel Gonzalez- Bocco for manuscript editorial support. Finally, we thank the medical and scientific staff of the Banco Municipal de Sangre and all of the patients and families who participated in the study.

## Author Contributions

**Conceptualization:** Anuraj H. Shankar, Elena N. Naumova, Irene Bosch.

**Data curation:** Baptiste Vasey, Bobby Brooke Herrera, Aniuska Becerra, Kris Xhaja, Elena N. Naumova.

**Formal analysis:** Baptiste Vasey, Anuraj H. Shankar, Bobby Brooke Herrera, Aniuska Becerra, Kris Xhaja, Diana Caicedo, John Miller, Elena N. Naumova, Irene Bosch.

**Funding acquisition:** Irene Bosch.

**Investigation:** Bobby Brooke Herrera, Aniuska Becerra, Kris Xhaja, Marion Echenagucia, Sara R. Machado, Elena N. Naumova, Irene Bosch.

**Methodology:** Anuraj H. Shankar, Bobby Brooke Herrera, Aniuska Becerra, Kris Xhaja, Marion Echenagucia, Sara R. Machado, Diana Caicedo, Elena N. Naumova, Irene Bosch.

**Project administration:** Aniuska Becerra, Irene Bosch.

**Resources:** Irene Bosch.

**Software:** Baptiste Vasey, Anuraj H. Shankar, John Miller, Elena N. Naumova.

**Supervision:** Anuraj H. Shankar, Bobby Brooke Herrera, Elena N. Naumova, Irene Bosch.

**Validation:** Anuraj H. Shankar, Sara R. Machado, Irene Bosch.

**Visualization:** Baptiste Vasey, John Miller, Elena N. Naumova, Irene Bosch.

**Writing – original draft:** Baptiste Vasey, Anuraj H. Shankar, Bobby Brooke Herrera, Elena N. Naumova, Irene Bosch.

**Writing – review & editing:** Baptiste Vasey, Anuraj H. Shankar, Bobby Brooke Herrera, Sara R. Machado, Elena N. Naumova, Irene Bosch.

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
