## [Decision Letter · Decision Letter 0]

8 Jan 2020

Dear Dr. Bosch:

Thank you very much for submitting your manuscript "Multivariate time-series analysis of biomarkers from a dengue cohort offers new approaches for diagnosis and prognosis" (#PNTD-D-19-01624) for review by PLOS Neglected Tropical Diseases. Your manuscript was fully evaluated at the editorial level and by independent peer reviewers. The reviewers appreciated the attention to an important problem, but raised some substantial concerns about the manuscript as it currently stands. These issues must be addressed before we would be willing to consider a revised version of your study. We cannot, of course, promise publication at that time.

We therefore ask you to modify the manuscript according to the review recommendations before we can consider your manuscript for acceptance. Your revisions should address the specific points made by each reviewer. 

When you are ready to resubmit, please be prepared to upload the following:

(1) A letter containing a detailed list of your responses to the review comments and a description of the changes you have made in the manuscript.

(2) Two versions of the manuscript: one with either highlights or tracked changes denoting where the text has been changed (uploaded as a "Revised Article with Changes Highlighted" file); the other a clean version (uploaded as the article file).

(3) If available, a striking still image (a new image if one is available or an existing one from within your manuscript). If your manuscript is accepted for publication, this image may be featured on our website. Images should ideally be high resolution, eye-catching, single panel images; where one is available, please use 'add file' at the time of resubmission and select 'striking image' as the file type. 

Please provide a short caption, including credits, uploaded as a separate "Other" file. If your image is from someone other than yourself, please ensure that the artist has read and agreed to the terms and conditions of the Creative Commons Attribution License at http://journals.plos.org/plosntds/s/content-license (NOTE: we cannot publish copyrighted images). 

(4) If applicable, we encourage you to add a list of accession numbers/ID numbers for genes and proteins mentioned in the text (these should be listed as a paragraph at the end of the manuscript). You can supply accession numbers for any database, so long as the database is publicly accessible and stable. Examples include LocusLink and SwissProt.

(5) To enhance the reproducibility of your results, we recommend that you deposit your laboratory protocols in protocols.io, where a protocol can be assigned its own identifier (DOI) such that it can be cited independently in the future. For instructions see http://journals.plos.org/plosntds/s/submission-guidelines#loc-methods

While revising your submission, please upload your figure files to the Preflight Analysis and Conversion Engine (PACE) digital diagnostic tool, https://pacev2.apexcovantage.com/ PACE helps ensure that figures meet PLOS requirements. To use PACE, you must first register as a user. Then, login and navigate to the UPLOAD tab, where you will find detailed instructions on how to use the tool. If you encounter any issues or have any questions when using PACE, please email us at figures@plos.org.

We hope to receive your revised manuscript by Mar 08 2020 11:59PM. If you anticipate any delay in its return, we ask that you let us know the expected resubmission date by replying to this email.

To submit a revision, go to https://www.editorialmanager.com/pntd/ and log in as an Author. You will see a menu item call Submission Needing Revision. You will find your submission record there. 

Sincerely,

Abdallah M. Samy, PhD

Guest Editor

Robert Reiner

Deputy Editor

Editor comments to authors: I invited and received three reviews of your manuscript. All reviews raised some important points to be addressed before considering a revised version of your manuscript. I would kindly ask you to address carefully all of their concerns before submitting your revised manuscript. Thanks, AMS

Reviewer's Responses to Questions

**Key Review Criteria Required for Acceptance?**

**Methods**

-Are the objectives of the study clearly articulated with a clear testable hypothesis stated?

-Is the study design appropriate to address the stated objectives?

-Is the population clearly described and appropriate for the hypothesis being tested?

-Is the sample size sufficient to ensure adequate power to address the hypothesis being tested?

-Were correct statistical analysis used to support conclusions?

-Are there concerns about ethical or regulatory requirements being met?

Reviewer #1: Yes (see comments below)

Reviewer #2: • Are failry clear, concise. 

• Nice description of the developed statistical analyses in R

• Human ethics number required? 

• L 223: 31 out of how many samples yielded suitable genome sequences? 

• Some manufacturer’s details are missing (e.g. Dengue IgM, IgG reagents, antibodies etc) 

• Since most statistical results are provided, the authors could include exact p values rather than just general significance.

Reviewer #3: One IRB approval is listed but there are several affiliations of authors. Was IRB approval sought at other study sites? 

The visit schedule is not clear in the “Data collection” section of the methods. 

For the genome analysis, it is not clear what the mutations were in comparison to? Was a reference sequence used or was this observed temporally during the outbreak? Please deposit the sequences at the time of publication (not only supplemental but in a repository).

**Results**

-Does the analysis presented match the analysis plan?

-Are the results clearly and completely presented?

-Are the figures (Tables, Images) of sufficient quality for clarity?

Reviewer #1: Yes

Reviewer #2: • Whilst providing statistical significance e.g. by * p<0.01 is commonly performed, stating exact p values are more suited here (include in figures). 

• All figures should include in figure legend or better on y-axes what is depicted, e.g. mean ± 95% CI. Please include this info in all relevant figures. Supp files have often ‘median’ in the figure legend, which is acceptable if no ranges are given. 

• Table 1: age sign >=7 should be ≥; 

• L290 states 61% were female, whereas Table 1 depicts 8 females in total in the study population. Please clarify. 

• Table 2: plateletes = typo, correct to platelets. 

• Figure 1: include number of subjects in brackets under each criteria/condition. Change notes * to another symbol to avoid confusion with statistical significance symbol (*).

• Figure 4A needs a description of what total count is depicted: mean? Median? Please include. 

• Sup Fig 6A needs to be increased as not legible at 400% zoom

• Abbreviations: check throughout, e.g. use MCP-1, use vWF in figures if used abbreviation in text

• L149-153, why were not some other factors tested that are known in Dengue pathobiology? E.g. MCP-1, TNFa, IFy, etc? Include a comment in results or discussion. It would have been nice to confirm findings from other researchers.

Reviewer #3: The abstract makes a nice distinction between the time points where these factors were significantly increased/decreased (e.g. at defervescence) but this is not clear from the results section. 

Lines 312-13 Provide the time point? The abstract indicates at defervescence? Provide the data for sST2 that was not significant? 

Figure 2, can plots be provided or absolute values (potentially as a supplemental data set)? The heat map is not specific enough and is difficult to read. Also, the interpretation of differences between significance for the kinetic and intercept are not described adequately or provided context for the broader audience. 

Figure 3 legend does not indicate what is meant by Day 0. Fever? Recruitment? Defervescence? Were there other clinical parameters that were collected that were not significant? 

For the clinical parameters provided in Fig. 2-3 can the AUC values be provided/discussed?

**Conclusions**

-Are the conclusions supported by the data presented?

-Are the limitations of analysis clearly described?

-Do the authors discuss how these data can be helpful to advance our understanding of the topic under study?

-Is public health relevance addressed?

Reviewer #1: A shortcoming, which may be attributed to the nature of the study design, is the lack of a control group. Of the 204 patients in the study population, 55 were excluded because they did not present with fever and one or more typical symptom, febrile illness for less than two days and no other obvious infection. It would be interesting, for future studies to also follow the progression of individuals who are asymptomatic but test positive for the presence of dengue RNA. This would enhance the scope of the study to also elucidate which biomarkers are of importance in the symptomatic developments of dengue fever at early timepoints in which individuals are viremic but asymptomatic and their subsequent progression to severe dengue.

Reviewer #2: ok

Reviewer #3: The abstract does not make it clear whether the transitions observed in genomes are reported as a marker of severity or some other association.

Although the intro and discussion adequately discuss the potential of clinical measures to predict SD, there is no acknowledgment of the development of biomarkers for SD prediction. 

Line 419- What do the authors consider a “hemostatic factor”. Also the use of the terms “hemodynamically stable” versus “unstable” are used throughout the discussion but how the authors define this is unclear. 

Line 463, this data is not included in the results and should be.

The title suggests that new approaches to diagnosis/prognosis are revealed by the data, but that is not fully clear since many of the variables used for this analysis have been considered by other groups and are either not consistent from cohort to cohort or no protocol exists for converting these values into a prognostic strategy.

**Editorial and Data Presentation Modifications?**

Reviewer #1: Minor revisions are suggested, since the nature of the study does not allow for further modifications

Reviewer #2: some figures should be increased in size and font size

figures (graphs) should include what is depicted on the y-axis: mean +/- SD or alike 

Accept with minor revisions

Reviewer #3: Specifically Figure 2 doesn't contain any raw data, only a heat map that was generated by the authors and not a statistical program regarding p-values. This needs to be improved and some representation of data itself included.

**Summary and General Comments**

Reviewer #1: The manuscript by Vasey presents unique insights into human biomarkers that can support current diagnostic tools in the effort to differentiate between dengue fever and similar febrile illnesses. The main strength of the study is its ability to differentiate dengue fever from other febrile illnesses. In terms of providing predictive approaches for the progression of dengue fever to severe dengue, more characterization would be needed to support the claim that as is, this study could aid as a predictive tool for the progression to severe dengue.

Reviewer #2: It is an interesting study. I would have liked to see some other markers tested in this study to link it to other existing studies ( e.g. TNFa, IFy levels). A major lack is the assessment of other, underlying medical conditions that may/could have masked/increased the symptoms. E.g. a full blood analysis, and viral genome testing by RT-PCR could have been beneficial.

 In summary, this manuscript is acceptable for publication upon completion of the suggested edits.

Reviewer #3: Multiple statistical methods were used to identify diagnostic/prognostic factors for dengue (vs. OFI) and mild vs. severe dengue in a cohort of suspected dengue patients from Nicaragua. The study describes a unique cohort that will be informative for the literature. Some of the analysis or results presentation was unclear and additional information are needed before this study is ready for publication.

The abstract does not make it clear whether the transitions observed in genomes are reported as a marker of severity or some other association.

PLOS authors have the option to publish the peer review history of their article (what does this mean?). If published, this will include your full peer review and any attached files.

Reviewer #1: No

Reviewer #2: No

Reviewer #3: No

---

## [Decision Letter · Decision Letter 1]

5 Mar 2020

Dear Dr. Bosch,

We are pleased to inform you that your manuscript 'Multivariate time-series analysis of biomarkers from a dengue cohort offers new approaches for diagnosis and prognosis' has been provisionally accepted for publication in PLOS Neglected Tropical Diseases.

Best regards,

Abdallah M. Samy, PhD

Deputy Editor

Robert Reiner

Deputy Editor

---

## [Editor Report · Acceptance letter]

10 Jun 2020

Dear Dr. Bosch,

We are delighted to inform you that your manuscript, "Multivariate time-series analysis of biomarkers from a dengue cohort offers new approaches for diagnosis and prognosis," has been formally accepted for publication in PLOS Neglected Tropical Diseases.

Best regards,

Shaden Kamhawi

co-Editor-in-Chief

Paul Brindley

co-Editor-in-Chief
